# Examining the Ability of CMIP6 Models to Reproduce the Upwelling SST Imprint in the Eastern Boundary Upwelling Systems

**Rubén Varela** [1,2,*]**, Maite DeCastro** [1,2]**, Laura Rodriguez-Diaz** [2]**, João Miguel Dias** [1] **and Moncho Gómez-Gesteira** [1,2]

[1] CESAM—Centre for Environmental and Marine Studies, Department of Physics, University of Aveiro, 3810-193 Aveiro, Portugal

[2] EphysLab—Environmental Physics Laboratory, CIM-UVIGO, Universidade de Vigo, Edificio Campus da Auga, 32004 Ourense, Spain

**\*** Correspondence: ruvarela@uvigo.es

**Abstract:** Knowing future changes in the sea surface temperature (SST) is of vital importance since they can affect marine ecosystems, especially in areas of high productivity such as the Eastern Boundary Upwelling Systems (EBUS). In this sense, it is key to have fine resolution models to study the SST patterns as close as possible to the coast where the upwelling influence is greater. Thus, the main objective of the present work is to assess the ability of 23 General Circulation Models (GCMs) from phase six of the Coupled Model Intercomparison Project (CMIP6) in reproducing the upwelling SST imprint in the EBUS through a comparison with the Optimum Interpolation of Sea Surface Temperature (OISST ¼) database of the National Oceanic and Atmospheric Administration for the common period of 1982–2014. The results have shown that most of the CMIP6 GCMs over-estimate nearshore SST for all the EBUS with the exception of Canary. Overall, the models with better resolution showed lower Normalized Root Mean Squared Error (NRMSE) and Normalized Bias (NBias), although the ability of the models is dependent on the study area. Thus, the most suitable models for each EBUS are the CNRM-HR, GFDL-CM4, HadGEM-MM, CMCC-VHR4, and EC-Earth3P for Canary; CESM1-HR, CMCC-VHR4, ECMWF-HR, and HadGEM-HM for Humboldt; and HadGEM-HH and HadGEM-HM for California. In the case of Benguela, no model adequately reproduces the SST imprint under the conditions established in the present study.

**Keywords:** SST; CMIP6 GCMs; OISST ¼, eastern boundary upwelling systems

## 1. Introduction

The sea surface temperature (SST) is a key variable in the study of parameters that affect ecosystems, especially in a climate change scenario. Reliable high-resolution SST data are especially important nearshore to capture coastal processes, such as the SST imprint of upwelling. In this sense, it is of vital importance to have databases that allow the correct analysis of the variability of the SST. Remote sensing products have been shown to perform well over the years [1,2]. In fact, one of the most widely used databases for reproducing historical SST patterns is the National Oceanic and Atmospheric Administration Optimum Interpolation Sea Surface Temperature (NOAA OISST ¼) database, with a resolution of 0.25° from 1982 to the present [3–8]. However, there is some controversy about the accuracy of remote sensing products when compared with in situ data [9]. Meneghesso et al. (2020) [9] found that Level 4 High Resolution Sea Surface Temperature (GHRSST) products can cause an overestimation of coastal SST and, thus, an underestimation of the thermal imprint of upwelling. Moreover, Dufois et al. (2012) [10] found warm bias in the monthly Pathfinder data during summer in the Eastern Boundary Upwelling Systems (EBUS), where high SST gradients exist.

In recent years, several authors have used projected SST data from General Circulation Models (GCMs) to assess future changes in marine heatwaves [11,12], climatic extremes [13], or upwelling [14,15]. In particular, many studies have focused on the SST changes that the main EBUS will undergo in the future under different climate change scenarios due to the influence of upwelling [15–18]. Data from GCMs of the Coupled Model Intercomparison Project Phase 5 (CMIP5) have been mostly used for this purpose [19], although they present some uncertainties when reproducing the coastal upwelling features. First, the coarse spatial resolution of GCMs (greater than 1°) may not capture special features nearshore, such as upwelling filaments [20–22]. This is especially patent in the vicinity of the coast where the strongest upwelling SST imprint can be found. Second, CMIP5 exhibits warm SST biases in EBUS [23–25], making it difficult to realistically reproduce the influence of upwelling on SST. Therefore, an update of the GCMs resolution and accuracy is essential to improve the ability to reproduce coastal temperature patterns in areas affected by upwelling [26,27]. These data are also crucial to drive dynamic downscaling approaches to analyze the impact of climate change on different species at regional scale [28–30].

In this sense, the Coupled Model Intercomparison Project Phase 6 (CMIP6) has recently been launched with the aim of improving the understanding of the physical processes, upgrading, among other things, the spatial resolution with models that reach up to 10 km [31]. The efficiency of CMIP6 to reproduce SST values has been recently tested by different authors. Richter and Tokinaga (2020) [32] evaluated the performance of CMIP6 GCMs in the tropical Atlantic. They found lower mean biases than for CMIP5, with the limitation of having few models available. Li et al. (2020) [33] compared the ability of CMIP5 and CMIP6 to simulate surface wind stress and SST over Tropical and Subtropical Oceans. They observed weaker upwelling-favorable winds using CMIP6 GCMs which may affect the SST imprint in the upwelling systems. Halder et al. (2021) [34] assessed the capacity of CMIP6 GCMs to reproduce the Tropical Indian Ocean (TIO) SST. They obtained good results for the inter-annual and decadal variability of TIO SST with an underestimation of the amplitude of variability.

The analysis of CMIP6 GCMs ability to reproduce the SST patterns nearshore is crucial since coastal marine biodiversity is closely linked to the colder SST imprint of EBUS, which usually takes place in the first 100 km from the coast [8]. For that purpose, it is extremely important to take advantage of the recent upgrade of the spatial resolution of the CMIP6 GCMs (up to 10 km), which allows the study of SST as close as possible to the coast. Despite the studies above mentioned, to date the adequacy of CMIP6 SST data to represent the particular characteristics of the SST in the EBUS has not been systematically analyzed. Only Varela et al. (2022) [15] studied coastal SST warming for the Canary Upwelling System using 6 CMIP6 GCMs. In their study, the comparison of historical SST values from CMIP6 GCMs and the OISST ¼ database showed a good agreement. However, the work was only focused on the Canary Upwelling System, leaving the rest of the main EBUS unexplored.

Therefore, the main objective of this work is to fill the existing knowledge gap by studying the adequacy of the CMIP6 GCMs to reproduce SST patterns in the main EBUS for the period of 1982–2014. For this purpose, SST data from 23 available CMIP6 GCMs will be compared with that from OISST ¼ database for the Canary, Benguela, Humboldt, and California Upwelling Systems (CUS, BUS, HUS, and CAUS, from now on). This analysis will assess the skill of CMIP6 GCMs to identify the SST patterns in the EBUS and can be used as a guide to select those models that best reproduce the particular behaviors of each area for future studies.

## 2. Methods

### 2.1. SST Data

Daily SST data were retrieved from the National Oceanic and Atmospheric Administration Optimum Interpolation Sea Surface Temperature database (NOAA OISST ¼; https://www.ncdc.noaa.gov/oisst (accessed on 30 January 2022)) at a resolution of 0.25°. Daily SST values were averaged at a monthly scale.

Monthly SST values were retrieved from 23 GCMs available within the framework of the CMIP6 project including the High Resolution Model Intercomparison Project (HighResMIP) (accessed on 30 January 2022) for the 1982–2014 period [31,35], which is common with OISST ¼. As some CMIP6 GCMs have different horizontal resolutions, a bilinear interpolation was carried out to convert SST data to a common 0.25° × 0.25° grid to conduct the comparison with OISST ¼. Table 1 shows a detailed description of the models used for this study. All the information in Table 1 was obtained from https://wcrp-cmip.github.io/CMIP6_CVs/docs/CMIP6_source_id.html (accessed on 1 November 2022). The NICAM16-8S is shown as a control model due to its special characteristics [36]. Two different experiments were considered: (i) Historical and (ii) Hist-1950. Both are siblings but, while Historical experiments start in 1850, Hist-1950 starts in 1950. Moreover, Hist-1950 simulations are at high and standard resolutions with a minimum atmosphere of 25–50 km at mid-latitudes, an ocean resolution of 0.25 degrees, and a minimum of daily coupling between ocean and atmosphere. More information about both experiments can be found at: https://es-doc.org/ (accessed on 1 November 2022).

**Table 1.** List of the Global Climate Models (GCMs) from the CMIP6 project (https://www.wcrp-climate.org/wgcm-cmip/wgcm-cmip6 (accessed on 1 November 2022)).

| Model Number | Name | Experiment ID | Oceanic Resolution (°) | Atmospheric Resolution (°) | Variant Label |
|---|---|---|---|---|---|
| 1 | AWI-CM-1-1-MR | Historical | 0.25 | 1 | r1i1p1f1 |
| 2 | CMCC-CM2-HR4 | Historical | 0.25 | 1 | r1i1p1f1 |
| 3 | CNRM-CM6-1-HR | Historical | 0.25 | 1 | r1i1p1f2 |
| 4 | GFDL-CM4 | Historical | 0.25 | 1 | r1i1p1f1 |
| 5 | GFDL-ESM4 | Historical | 0.5 | 1 | r1i1p1f1 |
| 6 | HadGEM3-GC31-MM | Historical | 0.25 | 1 | r1i1p1f3 |
| 7 | ICON-ESM-LR | Historical | 0.5 | 2.5 | r1i1p1f1 |
| 8 | MPI-ESM1-2-HR | Historical | 0.5 | 1 | r1i1p1f1 |
| 9 | BCC-CSM2-HR | Hist-1950 | 0.5 | 0.5 | r1i1p1f1 |
| 10 | CESM1-CAM5-SE-HR | Hist-1950 | 0.1 | 0.25 | r1i1p1f1 |
| 11 | CMCC-CM2-HR4 | Hist-1950 | 0.25 | 1 | r1i1p1f1 |
| 12 | CMCC-CM2-VHR4 | Hist-1950 | 0.25 | 0.25 | r1i1p1f1 |
| 13 | CNRM-CM6-1-HR | Hist-1950 | 0.25 | 1 | r1i1p1f2 |
| 14 | EC-Earth3P | Hist-1950 | 1 | 0.8 | r3i1p2f1 |
| 15 | EC-Earth3P-HR | Hist-1950 | 0.25 | 0.5 | r1i1p2f1 |
| 16 | ECMWF-IFS-HR | Hist-1950 | 0.25 | 0.25 | r1i1p1f1 |
| 17 | ECMWF-IFS-MR | Hist-1950 | 0.25 | 0.5 | r1i1p1f1 |
| 18 | FGOALS-f3-H | Hist-1950 | 0.1 | 0.25 | r1i1p1f1 |
| 19 | HadGEM3-GC31-HH | Hist-1950 | 0.1 | 0.5 | r1i1p1f1 |
| 20 | HadGEM3-GC31-HM | Hist-1950 | 0.25 | 0.5 | r1i1p1f1 |
| 21 | MPI-ESM1-2-HR | Hist-1950 | 0.5 | 1 | r1i1p1f1 |
| 22 | MPI-ESM1-2-XR | Hist-1950 | 0.5 | 0.5 | r1i1p1f1 |
| 23 | NICAM16-8S | HighresSST-present | None | 0.5 | r1i1p1f1 |

### 2.2. Analysis of Coastal and Oceanic SST

The area under study covers the main EBUS of the world (BUS, CUS, HUS, and CAUS, shown in Figure 1). Coastal points (blue) correspond to the grid points closest to the coast and oceanic points (red) to those that are 3 degrees away in the direction perpendicular to the coast. Thus, oceanic points are situated far enough from the coast to not be affected by upwelling [7,15]. The selection of both coastal and oceanic points was made following previous studies (Varela et al., 2018 [7], and the references therein).

Satellite and numerical data from each GCM were averaged over the entire period (1982–2014).

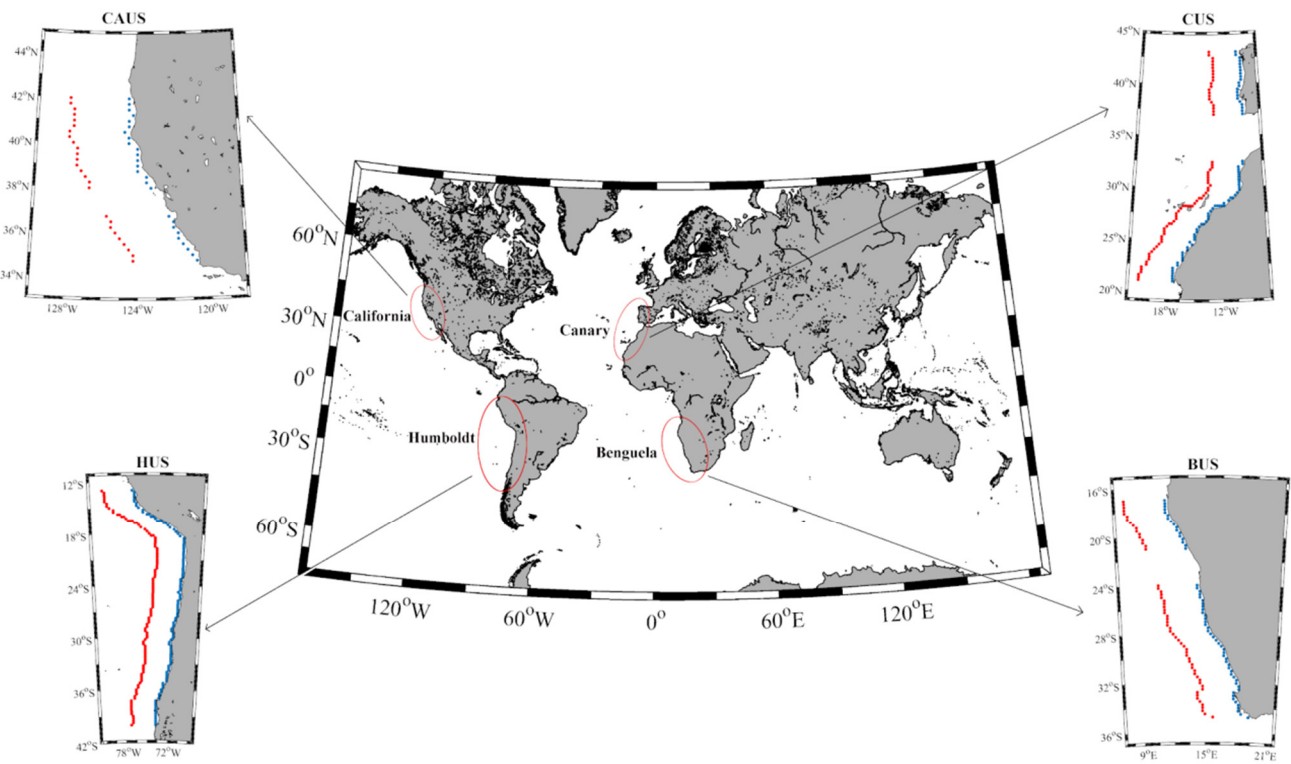

**Figure 1.** Eastern upwelling systems (EBUS). Blue (red) circles identify the coastal (oceanic) points selected. BUS: Benguela Upwelling System, CUS: Canary Upwelling System, HUS: Humboldt Upwelling System, and CAUS: California Upwelling System.

### 2.3. Validation

To assess the adequacy of each CMIP6 GCM to reproduce the SST values of OISST ¼, a validation process has been carried out by means of the normalized root mean square error (NRMSE) and the normalized bias error (NBias) [28,37,38]:

$$\text{NRMSE (\%)} = \frac{100}{\frac{1}{2}\left(\overline{\text{SST}_i^{\text{CMIP6}}} + \overline{\text{SST}_i^{\text{OISST}}}\right)} \cdot \sqrt{\frac{1}{N}\sum_{i=1}^{N}\left(\text{SST}_i^{\text{CMIP6}} - \text{SST}_i^{\text{OISST}}\right)^2} \quad (1)$$

$$\text{NBias (\%)} = \frac{100}{\frac{1}{2}\left(\overline{\text{SST}_i^{\text{CMIP6}}} + \overline{\text{SST}_i^{\text{OISST}}}\right)} \cdot \frac{1}{N}\sum_{i=1}^{N}\left(\text{SST}_i^{\text{CMIP6}} - \text{SST}_i^{\text{OISST}}\right) \quad (2)$$

where N is the total number of points in both data series, $\text{SST}_i^{\text{CMIP6}}$ refers to CMIP6 GCM values, and $\text{SST}_i^{\text{OISST}}$ denotes the values from OISST ¼. Barred variables correspond to mean values.

Thus, the smaller the NRSME and NBias values the better the ability of the CMIP6 GCM to reproduce the SST values provided by OISST ¼.

## 3. Results and Discussion

The SST maps from both the OISST ¼ database and the CMIP6 GCMs were represented for the BUS at an annual scale (Figure 2). The OISST ¼ map shows a clear gradient between coastal and oceanic temperature, with a lower coastal SST (between 14–16 °C) than in the open ocean (over 18 °C). Thus, a clear upwelling imprint is evident from 35° S to 18° S. This behavior had been previously observed by several authors. Santos et al. (2012a) [39] studied the differences between the coast and ocean from 1970 to 2009 using data from the UK Meteorological office, the Hadley Center HadISST1.1-Global Sea-Ice coverage, and SST, obtaining lower values for the coast than in open ocean (up to −5 °C). A similar pattern was found by Chen et al. (2012) [40] using MODIS Aqua daytime SST from 2003 to 2008. Regarding the CMIP6 GCMs, different behaviors can be observed. On the one hand, most models show the SST imprint of upwelling, although in many cases the cold coastal area is smaller than observed for OISST ¼. On the other hand, some models like CMCC-HR4 (historical and hist-1950) (2 and 11), ICON-LR (7), or EC-Earth3P (14) hardly show any trace of cold SST nearshore.

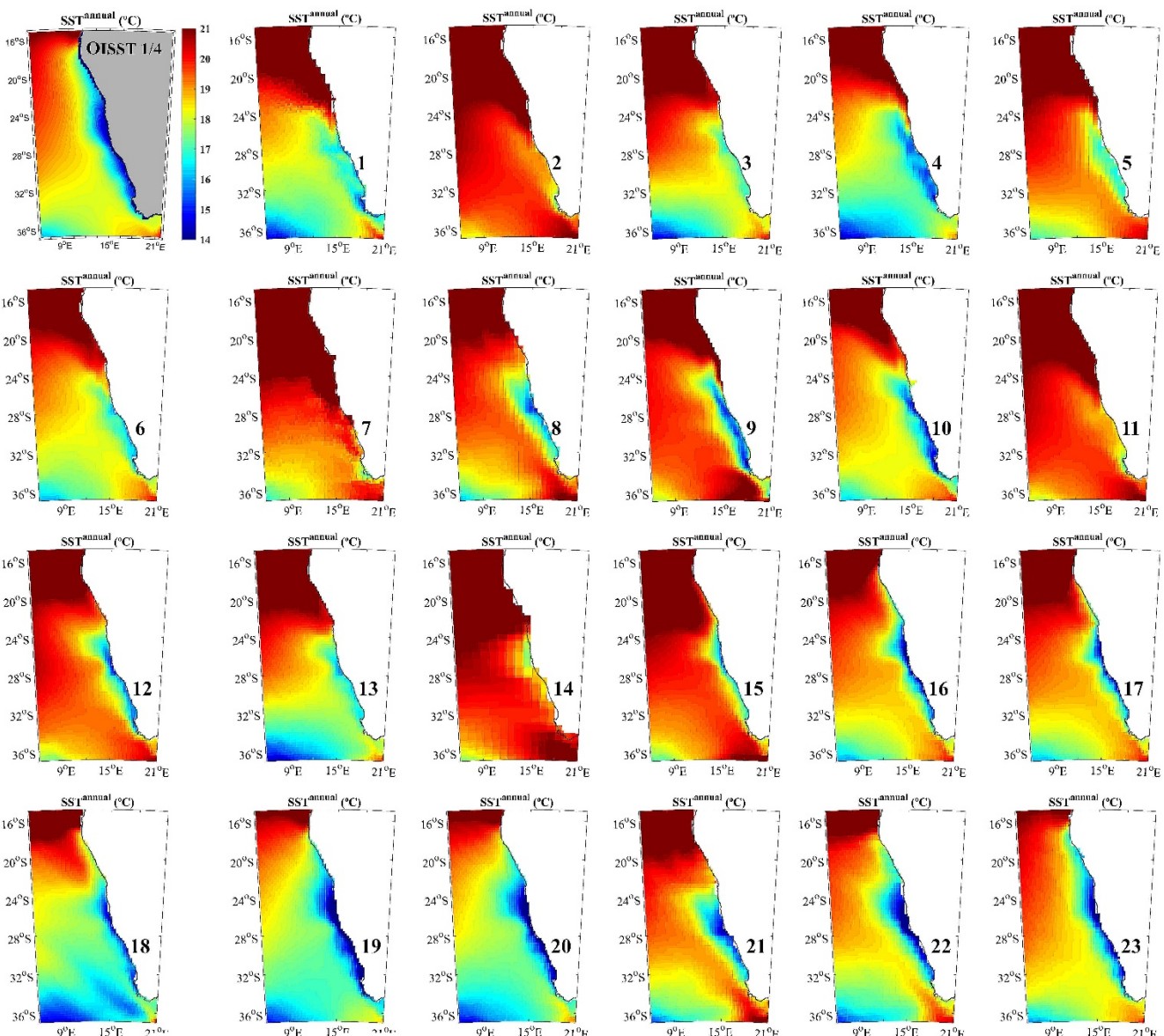

**Figure 2.** Mean of the annual SST (°C) for the Benguela upwelling system from 1982 to 2014. The numbers of each map refer to Table 1.



The ability of each CMIP6 GCM to reproduce the OISST ¼ SST is shown for the coastal (Figure 3a) and oceanic (Figure 3b) points previously depicted in Figure 1. It is evident from Figure 3a that most of the CMIP6 GCMs overestimate the coastal SST compared to OISST ¼, with some models exceeding satellite data by more than 4 °C. Only 7 GCMs adequately reproduce nearshore SST (ECMWF-HR and MR (16 and 17), FGOALS (18), HadGEM-HH and HM (19 and 20), and MPI-XR (22)). However, for the oceanic points, almost all of the CMIP6 GCMs show values similar (slightly higher) to those obtained for OISST ¼

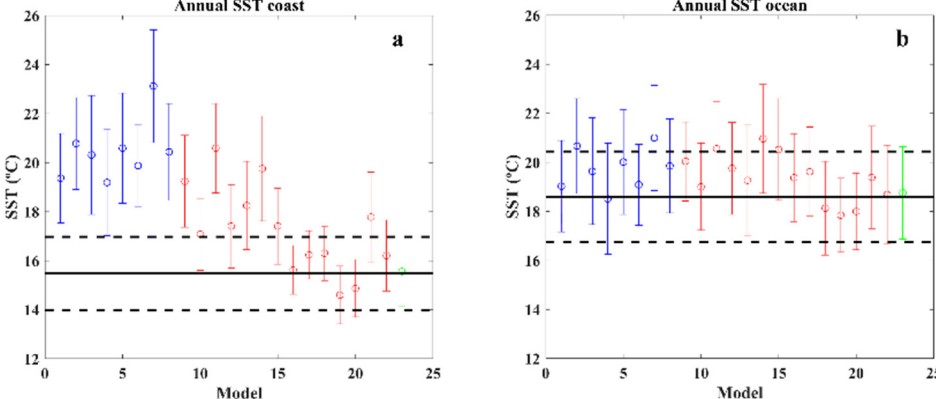

**Figure 3.** Benguela upwelling system mean SST value (±1 SD) for the OISST ¼ (black solid line) and each CMIP6 model for: (**a**) coastal and (**b**) oceanic points. The number and color of each model refer to Table 1.

The SST pattern for OISST ¼ and most of the CMIP6 GCMs is very similar for the CUS (Figure 4). In both cases, the influence of the upwelling on SST patterns is evident along the coastal zones, causing lower temperatures from 20° N to 34° N. Only the ICON-LR (7), BCC-HR (9), CESM1-HR (10), and FGOALS (18) barely show colder SST in the coast. Regarding the northernmost area of the CUS, situated in the western coast of the Iberian Peninsula, it does not show a clear upwelling influence on coastal SST due to the strong seasonal behavior which limits the effect of upwelling at the annual scale used for the figures [41–44].

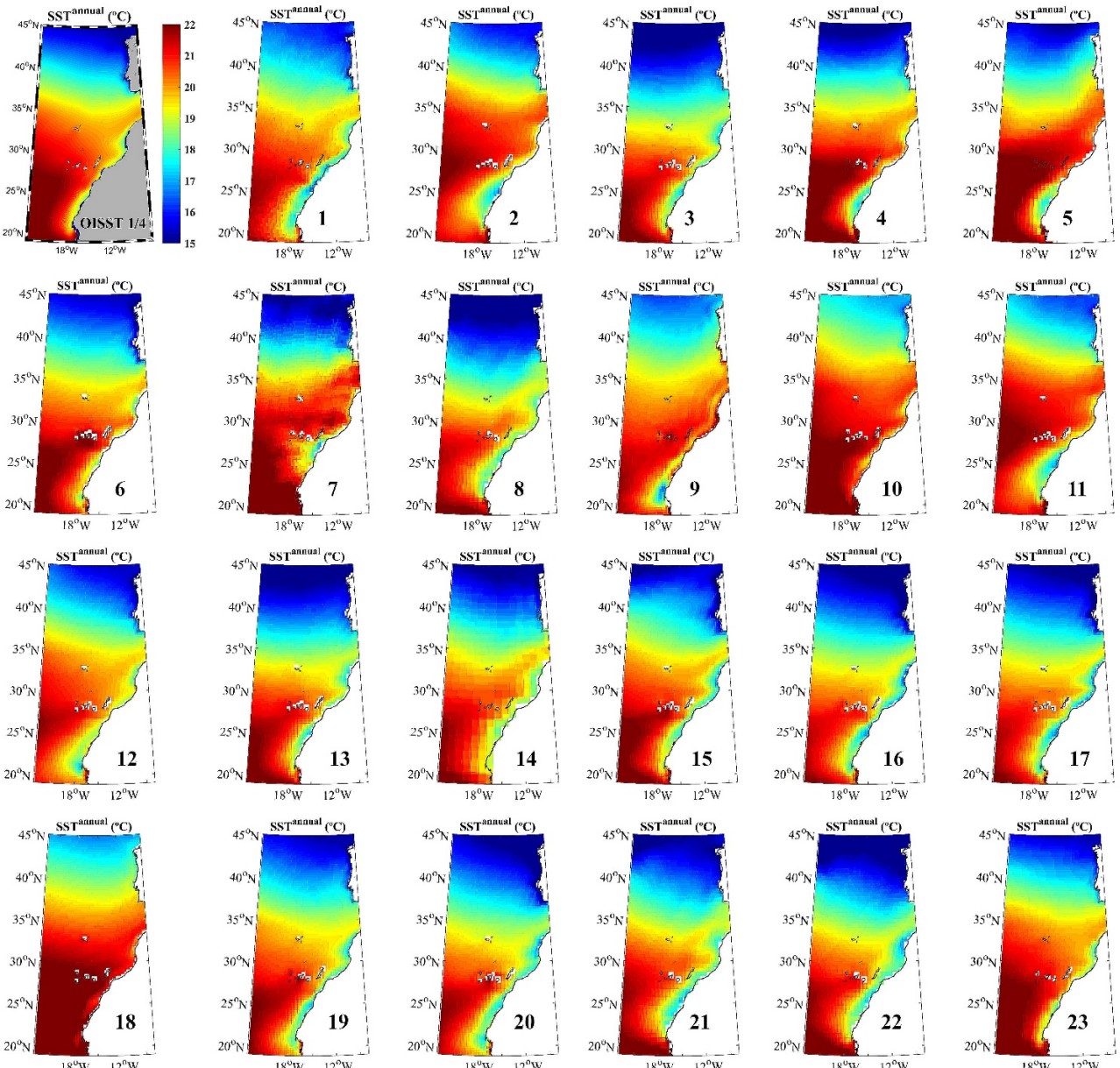

**Figure 4.** Mean of the annual SST (°C) for the Canary upwelling system from 1982 to 2014. The numbers of each map refer to Table 1.

Regarding the ability of each CMIP6 GCM to reproduce the SST values of OISST ¼, practically all the CMIP6 GCMs show similar SST values to OISST ¼ both for the coast (Figure 5a) and the ocean (Figure 5b), displaying a slight underestimation for most models.

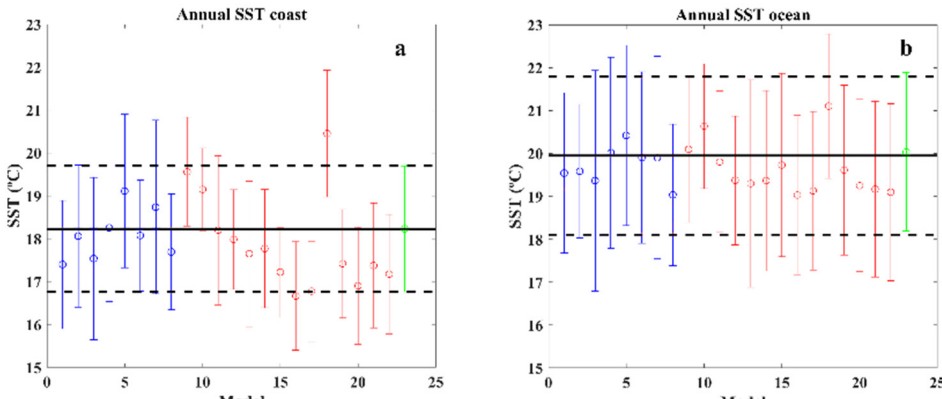

**Figure 5.** Canary upwelling system mean SST value (±1 SD) for the OISST ¼ (black solid line) and each CMIP6 model for: (**a**) coastal and (**b**) oceanic points. The number and color of each model refer to Table 1.

The HUS OISST ¼ map shows the influence of upwelling on SST in the southernmost part of the domain (~38° S), around 24° S in the area of Chile, and in Peru between 17° S and 13° S (Figure 6). Gutiérrez et al. (2011) [45] obtained similar results for the summer SST on the Peruvian coast from 1985 to 2005 using Pathfinder High Resolution data (~4 km). In general, most models present more difficulties to capture the cold SST signal along the coast than in the ocean for the Chile sub-region. However, the upwelling influence on the SST pattern is more visible for the area of Peru, although with less extension than for OISST ¼. Only a few CMIP6 GCMs (CMCC-HR4, historical and hist-1950 (2 and 11), and HadGEM-MM (6)) do not show the upwelling imprint in the area.

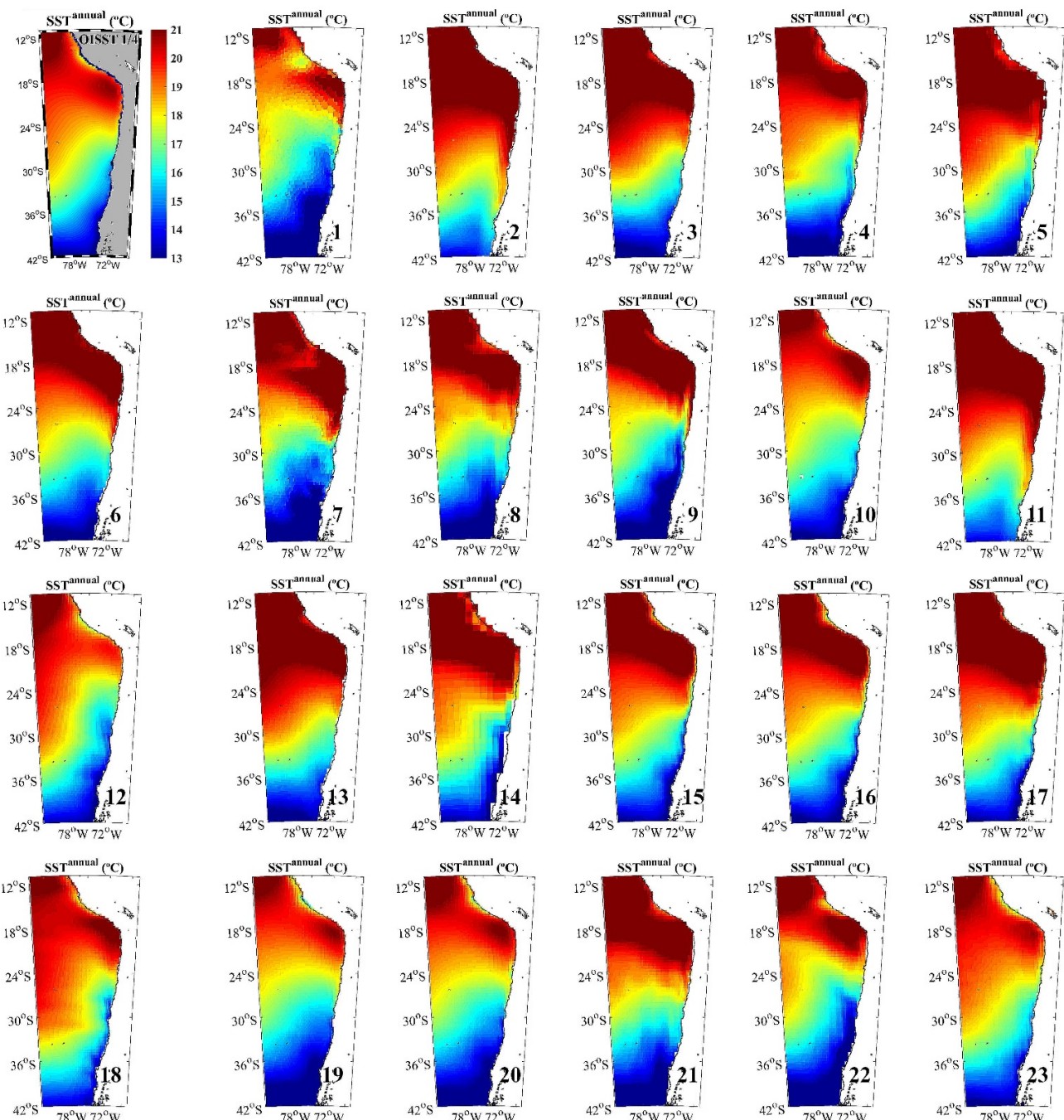

**Figure 6.** Mean of the annual SST (°C) for the Humboldt upwelling system from 1982 to 2014. The numbers of each map refer to Table 1.

As in the case of the CUS, almost all CMIP6 GCMs are able to reproduce the OISST ¼ values within the margin of error for both the coast and the ocean (Figure 7a, b), with a slight overestimation. Only CMCC-HR4 (historical and hist-1950) (2 and 11), HadGEM-MM (6), and ICON-LR (7) clearly overestimate the coastal SST with respect to OISST¼ by more than 2 °C.

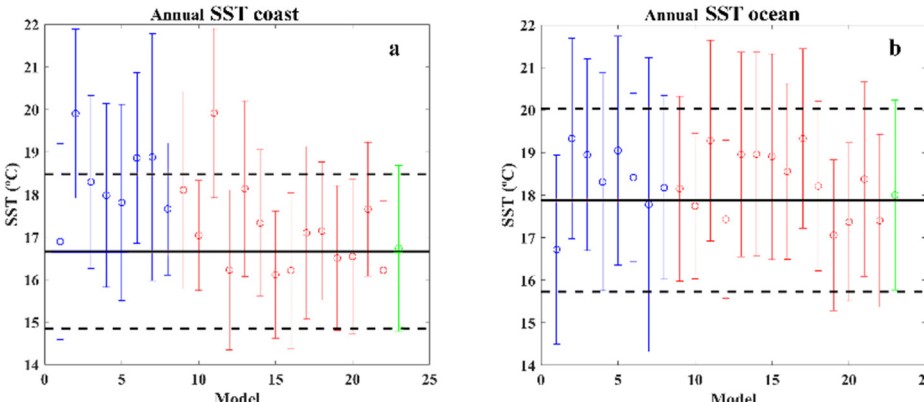

**Figure 7.** Humboldt upwelling system mean SST value (±1 SD) for the OISST ¼ (black solid line) and each CMIP6 model for: (**a**) coastal and (**b**) oceanic points. The number and color of each model refer to Table 1.

The CAUS OISST ¼ maps show lower SST along the coast than in the open ocean (Figure 8) as previously observed by different authors [8,46]. In the case of CMIP6 GCMs, different SST patterns can be observed. Most of them display lower SST values along the coast than in the ocean, occupying a less extensive region than provided by OISST ¼. In the case of CMCC-HR4 (historical and hist-1950) (2 and 11), CNRM-HR (historical and hist-1950) (3 and 13), and FGOALS (18), the influence of upwelling on SST is barely visible.

**Figure 8.** Mean of the annual SST (°C) for the California upwelling system from 1982 to 2014. The numbers of each map refer to Table 1.

Regarding the performance of the CMIP6 GCMs to reproduce the coastal and oceanic values provided by OISST ¼, almost all models overestimate coastal SST with half of them exceeding the error range of OISST ¼ (Figure 9a). However, oceanic values remain within acceptable margins compared to satellite data (Figure 9b).

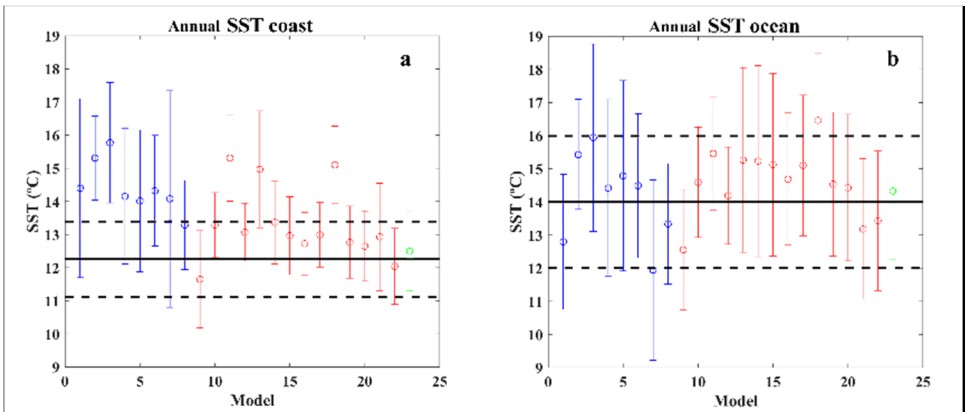

**Figure 9.** California upwelling system mean SST value (± 1 SD) for the OISST ¼ (black solid line) and each CMIP6 model for: (**a**) coastal and (**b**) oceanic points. The number and color of each model refer to Table 1.

Regarding the variability of the SST in terms of seasonality, in most cases no dependence has been observed depending on the season. This is caused by the existence of a quasi-permanent upwelling in all EBUS [47–50]. In the particular case of the CUS, the only season that shows some differences is summer (JAS) due to the seasonality of the upwelling in the northern area (west coast of the Iberian Peninsula). As the influence of upwelling on SST is more pronounced during the summer season in the north, practically all models are able to reproduce the coastal SST signal. On the other hand, the quasi-permanent upwelling in the south causes a visible SST imprint throughout the whole year [51]. A similar behavior is observed in CAUS, where those months corresponding to JFM show difficulties in reproducing the influence of upwelling on the SST both in OISST ¼ and in the CMIP6 GCMs. As in the case of the western Iberian Peninsula, although CAUS has a quasi-permanent upwelling, the least favorable conditions occur in winter (JFM), which means that a colder SST is barely observed on the coast than in the ocean [52]. The figures corresponding to the SST seasonal study can be consulted in the Supplementary Material.

The capability of each CMIP6 GCM to adequately reproduce the SST values of OISST ¼ is assessed by means of the NRSME and the NBias (see Equations (1) and (2)). This process allows selecting those models that best reproduce the SST values provided by OISST ¼ for each zone, both for the coast and ocean (Table 2 and Figure 10).

**Table 2.** NRMSE (%) and NBias (%) for each CMIP6 GCM at coastal and oceanic locations of each EBUS (see points in Figure 1). The number of each model refers to Table 1. Bold values indicate those models with NRMSE and NBIAS less than ±5%.

| | Benguela | | | | Canary | | | | Humboldt | | | | California | | | |
|---|---|---|---|---|---|---|---|---|---|---|---|---|---|---|---|---|
| | NRMSE (%) | | NBias (%) | | NRMSE (%) | | NBias (%) | | NRMSE (%) | | NBias (%) | | NRMSE (%) | | NBias (%) | |
| Model | Coast | Ocean | Coast | Ocean | Coast | Ocean | Coast | Ocean | Coast | Ocean | Coast | Ocean | Coast | Ocean | Coast | Ocean |
| 1 | 23.93 | 7.15 | 22.28 | 2.28 | 7.01 | 3.46 | −4.67 | −2.04 | 6.49 | 7.81 | 1.38 | −6.74 | 19.03 | 9.07 | 16.04 | −8.91 |
| 2 | 32.43 | 11.70 | 29.21 | 10.52 | 5.75 | 5.53 | −0.92 | −1.77 | 18.79 | 8.40 | 17.95 | 7.76 | 22.35 | 9.81 | 22.12 | 9.74 |
| 3 | 29.05 | 7.98 | 27.02 | 5.43 | **4.97** | **3.10** | **−3.86** | **−2.89** | 10.32 | 6.78 | 9.59 | 5.77 | 25.16 | 13.02 | 25.08 | 12.99 |
| 4 | 23.17 | 6.01 | 21.43 | −0.45 | **3.96** | **1.93** | **0.69** | **0.33** | 9.03 | 3.94 | 7.57 | 2.36 | 14.69 | 3.35 | 14.09 | 3.02 |
| 5 | 29.48 | 8.89 | 28.36 | 7.35 | 8.45 | 3.29 | 5.60 | 2.34 | 10.34 | 7.21 | 9.37 | 6.29 | 13.06 | 5.73 | 12.51 | 5.50 |

| | | | | | | | | | | | | | | | | |
|---|---|---|---|---|---|---|---|---|---|---|---|---|---|---|---|---|
| 6 | 25.92 | 4.99 | 24.86 | 2.60 | **4.14** | **1.89** | **−0.86** | **−0.15** | 14.40 | 5.30 | 12.61 | 2.89 | 15.77 | 3.65 | 15.51 | 3.46 |
| 7 | 43.89 | 14.75 | 39.17 | 12.12 | 7.74 | 3.38 | 3.42 | −0.24 | 15.48 | 8.08 | 11.34 | −0.59 | 19.55 | 17.10 | 14.60 | −15.82 |
| 8 | 28.96 | 7.37 | 27.62 | 6.52 | 4.33 | 5.24 | −2.97 | −4.69 | 6.89 | 6.39 | 5.80 | 1.62 | 9.96 | 5.28 | 8.06 | −4.82 |
| 9 | 30.13 | 8.55 | 21.64 | 7.45 | 11.00 | 3.51 | 7.34 | 0.74 | 12.68 | 8.83 | 8.11 | 1.47 | 7.87 | 11.47 | −5.09 | −10.85 |
| 10 | 16.96 | 4.38 | 9.81 | 2.17 | 6.01 | 3.99 | 4.98 | 3.38 | **4.04** | **1.52** | **2.22** | **−0.81** | 8.21 | 4.31 | 8.03 | 4.17 |
| 11 | 31.58 | 11.23 | 28.30 | 9.99 | 5.90 | 5.61 | −0.18 | −0.65 | 18.85 | 8.21 | 18.00 | 7.51 | 22.37 | 10.00 | 22.17 | 9.93 |
| 12 | 16.91 | 7.08 | 11.69 | 6.02 | **3.09** | **4.31** | **−1.36** | **−2.85** | **3.27** | **3.65** | **−2.41** | **−2.57** | 6.75 | 2.04 | 6.41 | 1.38 |
| 13 | 19.64 | 7.54 | 16.42 | 3.54 | **4.59** | **3.43** | **−3.24** | **−3.23** | 9.72 | 6.82 | 8.69 | 5.80 | 20.08 | 8.70 | 19.91 | 8.66 |
| 14 | 24.19 | 12.73 | 23.67 | 11.95 | **2.48** | **3.57** | **−1.31** | **−2.97** | 5.93 | 8.02 | 1.62 | 5.84 | 10.71 | 8.63 | 10.36 | 8.45 |
| 15 | 13.09 | 10.51 | 11.67 | 9.85 | 6.43 | 1.85 | −5.69 | −1.03 | 4.70 | 6.92 | −3.10 | 5.54 | 5.77 | 7.97 | 5.61 | 7.72 |
| 16 | 5.71 | 5.44 | 0.91 | 4.10 | 9.48 | 4.90 | −8.97 | −4.63 | **4.20** | **4.94** | **−2.50** | **3.69** | 4.55 | 5.11 | 3.75 | 4.83 |
| 17 | 8.56 | 6.97 | 4.78 | 5.34 | 8.85 | 4.38 | −8.31 | −4.10 | 5.15 | 9.31 | 2.83 | 7.76 | 6.34 | 7.63 | 5.80 | 7.57 |
| 18 | 6.61 | 6.40 | 4.64 | −2.55 | 11.91 | 5.68 | 11.54 | 5.62 | 5.59 | 3.56 | 2.64 | 1.83 | 20.81 | 16.23 | 20.77 | 16.17 |
| 19 | 8.58 | 4.69 | −5.87 | −4.13 | 5.96 | 2.49 | −4.55 | −1.72 | 4.10 | 5.48 | −0.93 | −4.74 | **4.24** | **3.98** | **4.08** | **3.79** |
| 20 | 8.84 | 4.22 | −4.01 | −3.28 | 8.27 | 4.16 | −7.55 | −3.48 | **4.32** | **4.14** | **−0.48** | **−2.93** | 3.58 | 3.66 | 3.16 | 3.08 |
| 21 | 16.92 | 5.81 | 12.85 | 4.15 | 6.22 | 4.60 | −4.80 | −3.99 | 7.28 | 7.62 | 5.78 | 2.71 | 8.90 | 6.42 | 5.31 | −5.98 |
| 22 | 8.88 | 2.94 | 3.55 | 0.46 | 6.72 | 4.77 | −5.94 | −4.37 | 7.80 | 5.68 | −2.71 | −2.74 | 3.95 | 5.79 | −1.79 | −4.15 |
| 23 | **2.27** | **0.94** | **0.16** | **0.83** | **1.64** | **0.79** | **0.41** | **0.59** | **1.65** | **1.33** | **−0.70** | **0.65** | **2.85** | **2.42** | **2.36** | **2.36** |

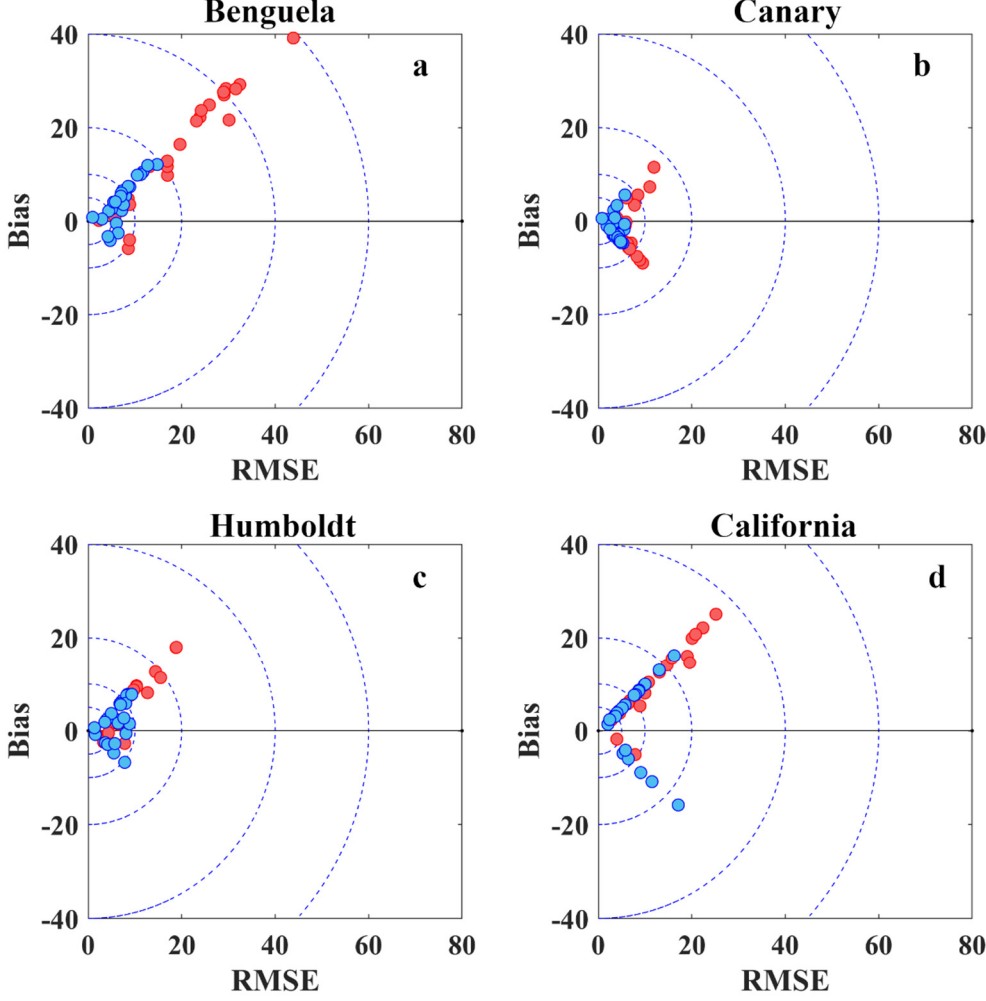

**Figure 10.** Graphic representation of values shown in Table 2. (**a**) Benguela, (**b**) Canary, (**c**) Humboldt, and (**d**) California. Red (blue) circles represent coastal (oceanic) locations.

From Table 2 and Figure 10 it is easy to observe that the NRMSE and NBias associated with the coastal locations are clearly higher than those for the oceanic locations for each upwelling system. Thus, these results evidence the difficulties of most of the CMIP6 GCMs to adequately reproduce the SST imprint caused by upwelling nearshore. Moreover, it is also evident that those models belonging to the hist-1950 experiment show lower

NRMSE and NBias than those from the historical experiment, probably due to the resolution improvement (Table 1). However, it is important to highlight the key differences that exist for the different upwelling zones. Among all the EBUS, the BUS seems to be where the CMIP6 GCMs show the greatest difficulties to reproduce the SST values from OISST ¼. Most models show NRMSE and NBias values greater than 10% and even 20%. On the other hand, the smallest NRMSE and NBias are observed for the CUS with almost all models within the interval ±10%. In particular, CNRM-HR (historical and hist-1950) (3 and 13), GFDL-CM4 (4), HadGEM-MM (6), CMCC-VHR4 (12), and EC-Earth3P (14) are the models that better reproduce SST values with NRMSE and NBias values within the interval ±5%. Regarding the HUS, moderate NRMSE and NBias, smaller than 10%, are obtained for the hist-1950 models while the historical models tend to exceed that value. The most suitable CMIP6 GCMs are CESM1-HR (10), CMCC-VHR4 (12), ECMWF-HR (16), and HadGEM-HM (20). Finally, the CAUS displays a behavior similar to BUS, with NRMSE and NBias values greater than 20%. The models that best reproduce SST values are HadGEM-HH and HadGEM-HM (19 and 20). In view of these results, taking into account the models resolution shown in Table 1, it is evident that the models with a finer resolution have lower NRMSE and NBias than the ones with a coarser resolution for all the EBUS except for CUS where almost all the models already adequately reproduce the influence of upwelling on SST.

Recently, different authors have evaluated the ability of various CMIP6 GCMs to reproduce SST. Sylla et al. (2022) [53] studied the impact of increased resolution to represent the CUS in climate models. They used a small sample of 6 HighResMIP also included in the present study. They found contradictory results depending on the area of the CUS. They concluded that increasing resolutions were not a sufficient condition to improve the influence of upwelling on the SST patterns. Farneti et al. (2022) [54] evaluated the biases in the CMIP6 GCMs focusing on the BUS. They found that biases remain in CMIP6 models but with an important reduction for those from the HighResMIP. Balaguru et al. (2021) [55] examined the influence of model resolution on coastal upwelling in the CAUS region for the Earth System Models (ESM). They found important coastal SST biases for the standard resolution models (1° atmosphere, 30–60 km ocean). They also observed an improvement of the nearshore SST biases for the high-resolution version of the models (0.25° atmosphere, 6–18 km ocean). Liu et al. (2022) [56] analyzed the performance of 48 CMIP6 models simulating the SST compared with observations from 1900 to 2014 by means of a multi-model ensemble. They obtained a clear SST overestimation for the CMIP6 GCMs with the highest biases for the BUS (up to 3 °C), and for HUS and CAUS (around 2 °C). The most modest biases were obtained in the case of CUS. Wang et al. (2022) [57] studied the seasonal SST extremes of 20 CMIP6 GCMs from 1981 to 2010 compared to World Ocean Atlas 2018 data. They found significant differences in seasonal SST biases in EBUS, especially in winter and summer, with the largest SST biases in BUS, HUS, and CAUS.

The difficulties of GCMs to reproduce SST patterns in upwelling regions have been an important topic of study already in previous phases of the CMIP. Richter and Xie (2008) [58] evaluated the origin of equatorial Atlantic biases in GCMs within the CMIP3 project from 1950 to 1999. They found simulated errors in the cross-equatorial winds and in the depth of the thermocline. In particular, they observed a high bias in the thermocline depth that prevented the appearance of an upwelling-related cold tongue in the area of South Africa. In this sense, the problems to reproduce realistic values for the equatorial thermocline and alongshore winds have been raised as the most important sources of bias in the replication of SST values in the upwelling areas [59]. Several authors have evaluated the Tropical Atlantic biases paying special attention to the BUS [27,60–62]. Most of these studies linked the warm bias of the SST to the need to improve the horizontal resolution to better capture wind patterns along the coast. In this sense, Ritcher and Tokinaga (2020) [32] conducted an overview of the performance of CMIP6 models in the tropical Atlantic. They compared the SST biases between CMIP phases 5 and 6 and found smaller biases for CMIP6 associated with stronger alongshore winds in the BUS. However, they also found

that the reduction in the SST bias was limited to summer (JJA). Even considering the improvement of CMIP6 GCMs over CMIP5, they cannot reproduce the upwelling pattern in the BUS. In addition, Song et al. (2020) [63] evaluated the differences in the eastern equatorial Pacific SST seasonal cycle between phases 5 and 6 of CMIP. That study includes the SST bias in the northern section of the HUS. They observed an improvement of the equatorial Pacific SST for CMIP6 over CMIP5, but still found a significant bias in the SST. In fact, the coastal area of Peru is one of the regions with the smallest improvements compared to CMIP5. Recently, Wang et al. (2022) [57] linked the largest SST biases in BUS, HUS, and CAUS to difficulties in reproducing the wind and cloud values. They concluded that the seasonal SST biases in the EBUS may be related to the difficulties to reproduce seasonal clouds and upwelling processes.

Our results are in line with those obtained by the previous authors. We observe an improvement in the ability to reproduce SST for the CMIP6 GCMs with higher resolution (hist-1950 experiment models) with respect to the low-resolution ones (historical models) both for coastal and oceanic locations. In fact, this improvement is visible for all EBUS with the exception of CUS, where most of models adequately reproduce the influence of upwelling on the SST patterns for both experiments independently of the resolution as Sylla et al. (2022) [53] found in their study. However, contrary to Wang et al. (2022) [57], we did not find important differences considering the seasonal performance of the CMIP6 GCMs under study. Both at an annual and seasonal scale, CMIP6 GCMs tend to overestimate coastal and oceanic SST with respect to OISST ¼, although this overestimation is much more pronounced nearshore, especially in the BUS and CAUS, as pointed out by previous authors [56]. This causes certain difficulties to adequately reproduce the coastal SST imprint of upwelling. Although, it is true that the recent improvements in the resolution seem to be helping to reduce the SST biases. It is also important to point out that, considering that OISST ¼ may be overestimating coastal SST and therefore underestimating the influence of upwelling on SST patterns [9], and that the SST obtained for CMIP6 GCMs are even higher than those for OISST ¼, it is clear that most CMIP6 GCMs greatly overestimates coastal SST.

As previously mentioned, the greatest influence of upwelling on the SST patterns in the EBUS occurs in the first 100 km [8]. Therefore, it is essential to have models with resolutions fine enough to be able to study these regions as close as possible to the coast. It is a known fact that EBUS can act as thermal refugia, reducing warming and even causing cooling trends in the areas affected by this mechanism [7]. Thus, nowadays, EBUS are an important focus in many productivity studies due to their important impact on marine productivity and fisheries [64–66].

## 4. Conclusions

The present work studies the ability of CMIP6 GCMs to reproduce SST patterns in the main EBUS. In this sense, data provided by 23 CMIP6 GCMs has been compared with data based on satellite measurements provided by OISST ¼ for the common period of 1982–2014. Most of the CMIP6 GCMs have shown difficulties in reproducing coastal SST values, displaying a clear overestimation for most of the EBUS with the exception of CUS. However, an improvement has also been observed in terms of the NRMSE and the NBias associated with an upgrade in model resolution. Even so, the ability of each model to reproduce the upwelling SST imprint on EBUS is strongly dependent on the area under study. Thus, the most suitable models for each EBUS are:

Benguela: No model adequately reproduces the SST imprint under the conditions established in the present study.

Canary: CNRM-HR (historical and hist-1950) (3 and 13), GFDL-CM4 (4), HadGEM-MM (6), CMCC-VHR4 (12), and EC-Earth3P (14).

Humboldt: CESM1-HR (10), CMCC-VHR4 (12), ECMWF-HR (16), and HadGEM-HM (20).

California: HadGEM-HH and HadGEM-HM (19 and 20).

The present work can provide upcoming researchers with a guide to know which of the available CMIP6 GCMs best reproduce SST patterns and, thus, serve as a basis for future studies.

**Supplementary Materials:** The following supporting information can be downloaded at: https://www.mdpi.com/article/10.3390/jmse10121970/s1, Figure S1: Mean of the seasonal SST for JFM (°C) for the Benguela upwelling system from 1982 to 2014. The numbers of each map refer to Table 1. Figure S2: Mean of the seasonal SST for AMJ (°C) for the Benguela upwelling system from 1982 to 2014. The numbers of each map refer to Table 1. Figure S3: Mean of the seasonal SST for JAS (°C) for the Benguela upwelling system from 1982 to 2014. The numbers of each map refer to Table 1. Figure S4: Mean of the seasonal SST for OND (°C) for the Benguela upwelling system from 1982 to 2014. The numbers of each map refer to Table 1. Figure S5: Benguela upwelling system seasonal mean SST value (± 1 SD) for the OISST ¼ (black solid line) and each CMIP6 model for coastal (left panels) and oceanic (right panels) points. The number and color of each model refer to Table 1. Figure S6: Mean of the seasonal SST for JFM (°C) for the Canary upwelling system from 1982 to 2014. The numbers of each map refer to Table 1. Figure S7: Mean of the seasonal SST for AMJ (°C) for the Canary upwelling system from 1982 to 2014. The numbers of each map refer to Table 1. Figure S8: Mean of the seasonal SST for JAS (°C) for the Canary upwelling system from 1982 to 2014. The numbers of each map refer to Table 1. Figure S9: Mean of the seasonal SST for OND (°C) for the Canary upwelling system from 1982 to 2014. The numbers of each map refer to Table 1. Figure S10: Canary upwelling system seasonal mean SST value (±1 SD) for the OISST ¼ (black solid line) and each CMIP6 model for coastal (left panels) and oceanic (right panels) points. The number and color of each model refer to Table 1. Figure S11: Mean of the seasonal SST for JFM (°C) for the Humboldt upwelling system from 1982 to 2014. The numbers of each map refer to Table 1. Figure S12: Mean of the seasonal SST for AMJ (°C) for the Humboldt upwelling system from 1982 to 2014. The numbers of each map refer to Table 1. Figure S13: Mean of the seasonal SST for JAS (°C) for the Humboldt upwelling system from 1982 to 2014. The numbers of each map refer to Table 1. Figure S14: Mean of the seasonal SST for OND (°C) for the Humboldt upwelling system from 1982 to 2014. The numbers of each map refer to Table 1. Figure S15: Humboldt upwelling system seasonal mean SST value (±1 SD) for the OISST ¼ (black solid line) and each CMIP6 model for coastal (left panels) and oceanic (right panels) points. The number and color of each model refer to Table 1. Figure S16: Mean of the seasonal SST for JFM (°C) for the California upwelling system from 1982 to 2014. The numbers of each map refer to Table 1. Figure S17: Mean of the seasonal SST for AMJ (°C) for the California upwelling system from 1982 to 2014. The numbers of each map refer to Table 1. Figure S18: Mean of the seasonal SST for JAS (°C) for the California upwelling system from 1982 to 2014. The numbers of each map refer to Table 1. Figure S19: Mean of the seasonal SST for OND (°C) for the California upwelling system from 1982 to 2014. The numbers of each map refer to Table 1. Figure S20: California upwelling system seasonal mean SST value (± 1 SD) for the OISST ¼ (black solid line) and each CMIP6 model for coastal (left panels) and oceanic (right panels) points. The number and color of each model refer to Table 1.

**Author Contributions:** R.V.: Conceptualization, Methodology, Formal analysis, Investigation, Writing—Original Draft. M.D.: Conceptualization, Methodology, Investigation, Writing—Review & Editing, Supervision. L.R.-D.: Conceptualization, Methodology, Writing—Review & Editing. J.M.D.: Writing—Review & Editing, Supervision. M.G.-G.: Conceptualization, Methodology, Writing—Review & Editing, Supervision. All authors have read and agreed to the published version of the manuscript.

**Funding:** This researcher was funded by Xunta de Galicia, Consellería de Cultura, Educación e Universidade, under Project ED431C 2021/44 "Programa de Consolidación e Estructuración de Unidades de Investigación Competitivas". Also by FCT/MCTES through the financial support to CESAM (UIDP/50017/2020+UIDB/50017/2020+LA/P/0094/2020) through national funds. Also, by project AquiMap (MAR-02.01.01-FEAMP-0022) co-financed by MAR2020 Program, Portugal 2020, and European Union though the European Maritime and Fisheries Fund. Also, forms part of the Marine Science programme (ThinkInAzul) supported by Ministerio de Ciencia e Innovación and Xunta de Galicia with funding from European Union NextGenerationEU (PRTR-C17.I1) and European Maritime and Fisheries Fund. Rubén Varela was supported by Xunta de Galicia through a post-doctoral grant (ED481B-2021-108)

**Institutional Review Board Statement:** Not applicable.

**Informed Consent Statement:** Not applicable.

**Data Availability Statement:** Data from this study are available in: https://www.ncdc.noaa.gov/oisst and https://esgf-node.llnl.gov/search/cmip6/ (accessed on 1 November 2022).

**Acknowledgments:** This work was partially financed by Xunta de Galicia, Consellería de Cultura, Educación e Universidade, under Project ED431C 2021/44 "Programa de Consolidación e Estructuración de Unidades de Investigación Competitivas". Rubén Varela was supported by Xunta de Galicia through a post-doctoral grant (ED481B-2021-108). Thanks are due to FCT/MCTES for the financial support to Centre for Environmental and Marine Studies (CESAM) (UIDP/50017/2020+UIDB/50017/2020), through national funds. Thanks also to project AquiMap (MAR-02.01.01-FEAMP-0022) co-financed by MAR2020 Program, Portugal 2020, and European Union though the European Maritime and Fisheries Fund. This study forms part of the Marine Science programme (ThinkInAzul) supported by Ministerio de Ciencia e Innovación and Xunta de Galicia with funding from European Union NextGenerationEU (PRTR-C17.I1) and European Maritime and Fisheries Fund.

**Conflicts of Interest:** The authors declare no conflict of interest.

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
