# Peer review of "Examining the Ability of CMIP6 Models to Reproduce the Upwelling SST Imprint in the Eastern Boundary Upwelling Systems"

_jmse, doi:10.3390/jmse10121970_

Round 1

Reviewer 1 Report

The present work aims to assess the ability of 16 23 General Circulation Models (GCMs) from the phase 6 of CMIP6 in reproducing the upwelling SST imprint in the EBUS through 18 a comparison with OISST ¼ . The topic is needed for model validation, however, the present work has some limitations:

(1) SST validation is usually done between model/satellite data and in situ data. This work lacks in situ data comparison, which is quite important for model result.

(2) The present work lacks discussion part. Why some products overestimate the upwelling and some not? Are there any ocean dynamic processes? Are the biases corelated to wind speed, solar radiation or some other factors? Please discuss. 

Author Response

Reviewer 1:

The present work aims to assess the ability of 23 General Circulation Models (GCMs) from the phase 6 of CMIP6 in reproducing the upwelling SST imprint in the EBUS through a comparison with OISST ¼ . The topic is needed for model validation, however, the present work has some limitations:

(1) SST validation is usually done between model/satellite data and in situ data. This work lacks in situ data comparison, which is quite important for model result.

NOAA Optimum Interpolation 1/4 Degree Daily Sea Surface Temperature (OISST) uses Advanced Very High Resolution Radiometer (AVHRR) satellite data from the Pathfinder AVHRR SST dataset when available for September 1981 through December 2005, and the operational Navy AVHRR Multi-Channel SST data for 2006 to the present day. The product also uses sea ice datasets, in situ data from ships and buoys, and includes a large-scale adjustment of satellite biases with respect to the in situ data. As mentioned by the NOAA, Pathfinder AVHRR SST was chosen because of good agreement with the in situ observation data (https://www.ncei.noaa.gov/access/metadata/landing-page/bin/iso?id=gov.noaa.ncdc:C00844).

Moreover, several authors have assessed the adequacy of OISST ¼ to reproduce in-situ SST data from buoys or loggers (Hughes et al., 2009; Brewin et al., 2017, 2018; Kwon et al., 2018; Fernandez-Novoa et al., 2020; Lee et al., 2020; Saracoglu et al., 2021; Stramska et al., 2021). However, we are aware of the limitations of the L4 GHRSST products, especially in the upwelling regions (Meneghesso et al., 2020) (Línes 55-59 and 376-381). Although it is true that for a study like the one presented, it is necessary to have a global and sufficiently long database, something that the in-situ data does not ensure, for these reasons we have decided to stick to the comparison between OISST ¼ and the CMIP6 GCMs.

(2) The present work lacks discussion part. Why some products overestimate the upwelling and some not? Are there any ocean dynamic processes? Are the biases correlated to wind speed, solar radiation or some other factors? Please discuss.

First, we want to clarify that across the study we are not evaluating upwelling, against, we are focusing on the influence of upwelling on SST. Although they may seem similar statements, the differences are tremendously important. Our study is not focused on the causes of upwelling. At no time do we assess the ability of the models to reproduce the upwelling in terms of wind stress, Ekman Transport, Ekman Pumping... Our interest lies solely in verifying whether the models are capable of reproducing the consequences of upwelling. That is, the influence of upwelling on coastal SST. As we mentioned throughout the text, the true interest of the EBUS lies in their ability to act as a thermal refuge (Seabra et al., 2019), which makes it an important focus in terms of biodiversity and productivity (Burrows et al., 2014; Lourenço et al., 2016; Renault et al., 2016). Therefore, the question that we ask ourselves and that we think is of interest is: can CMIP6 GCMs reproduce the SST imprint caused by upwelling? Because, after all, that is precisely the most important part if we want to conduct future studies of productivity, fisheries, fish migration...

Second, we agree with the reviewer that a discussion on the possible causes of the SST biases could be of interest. Thus, we have introduced a new paragraph (Lines 336-362) to discuss this topic.

References:

Hughes, S. L., Holliday, N. P., Colbourne, E., Ozhigin, V., Valdimarsson, H., Østerhus, S., & Wiltshire, K. (2009). Comparison of in situ time-series of temperature with gridded sea surface temperature datasets in the North Atlantic. ICES Journal of Marine Science, 66(7), 1467-1479.

Brewin, R. J., de Mora, L., Billson, O., Jackson, T., Russell, P., Brewin, T. G., ... & Fishwick, J. R. (2017). Evaluating operational AVHRR sea surface temperature data at the coastline using surfers. Estuarine, Coastal and Shelf Science, 196, 276-289.

Brewin, R. J., Smale, D. A., Moore, P. J., Dall’Olmo, G., Miller, P. I., Taylor, B. H., ... & Yang, M. (2018). Evaluating operational AVHRR sea surface temperature data at the coastline using benthic temperature loggers. Remote Sensing, 10(6), 925.

Kwon, K., Choi, B. J., & Lee, S. H. (2018). Assimilation of different SST datasets to a coastal ocean modeling system in the Yellow and East China Sea. Journal of Coastal Research, (85 (10085)), 1041-1045.

Fernández-Nóvoa, D., Costoya, X., Kobashi, D., Rodríguez-Díaz, L., deCastro, M., & Gómez-Gesteira, M. (2020). Influence of Mississippi and Atchafalaya River plume in the winter coastal cooling of the Northwestern Gulf of Mexico. Journal of Marine Systems, 209, 103374.

Lee, E. Y., & Park, K. A. (2020). Validation of satellite sea surface temperatures and long-term trends in Korean coastal regions over past decades (1982–2018). Remote Sensing, 12(22), 3742.

SaraçoÄŸlu, F. A., AydoÄŸan, B., Ayat, B., & SaraçoÄŸlu, K. E. (2021). Spatial and Seasonal Variability of Long-Term Sea Surface Temperature Trends in Aegean and Levantine Basins. Pure and Applied Geophysics, 178(9), 3769-3791.

Stramska, M., Konik, M., Aniskiewicz, P., Jakacki, J., & Darecki, M. (2021). Comparisons of Satellite and Modeled Surface Temperature and Chlorophyll Concentrations in the Baltic Sea with in Situ Data. Remote Sensing, 13(15), 3049.

Seabra, R., Varela, R., Santos, A. M., Gomez-Gesteira, M., Meneghesso, C., Wethey, D. S. and Lima, F. P. 2019. Reduced nearshore warming associated with eastern boundary upwelling systems. Frontiers in Marine Science, 6, 104.

Burrows, M. T., et al. (2014). Geographical limits to species-range shifts are suggested by climate velocity. Nature 507, 492–495.

Lourenço, C. R., et al. (2016). Upwelling areas as climate change refugia for the distribution and genetic diversity of a marine macroalga. J. Biogeogr. 43, 1595–1607.

Renault, L., Deutsch, C., McWilliams, J. C., Frenzel, H., Liang, J.-H., and Colas, F. (2016). Partial decoupling of primary productivity from upwelling in the California Current system. Nat. Geosci. 9, 505–508.

Reviewer 2 Report

This manuscript evaluated the simulated ability of the Eastern Boundary Upwelling System from 23 CMIP6 models, which is important for model result-users and model development. In general, the manuscript is well-structured and well-written. The method and conclusion are reliable. I think it can be accepted after clarifying several issues.

(a) As I know, the models participating in the CMIP6 are all general circulation models. Therefore, I do not think it’s necessary to include the “general circulation models” in the title. I suggest that the title can be changed to “On the ability of CMIP6 models to reproduce the upwelling SST imprint in the Eastern Boundary Upwelling Systems”. 

(b) It’s helpful if authors can give the meaning of “Historical”, “Hist-1950”, and “NICAM16-8S”. And I’m confused why the authors selected NICAM16-8S, which seems only an atmosphere model, as a control model. 

(c) As we know, there are several ensemble members in the CMIP6 published data, such as r1i1p1f1, r2i1p1f1, etc. In general, the climatological difference between r1 and r2 is very small, but it will be large between p1 and p2, or f1 and f2. The authors should clarify which ensemble members are used in this work. 

(d) It’ll be better if authors use the model’s name, even an abbreviation, instead of the number (such as 1, 2, 3, etc.). 

(e) This year, Liu et al. (doi:10.1016/j.dsr2.2022.105120) evaluated the global SST simulated ability from the 48 CMIP6 models and showed the ensemble mean biases. As I know, it’s the result of the most number of CMIP6 models. Authors can compare with figure 1 of Liu’s article, although it only gives a global distribution. 

(f) It’s not necessary to be answered, but it’s appreciated if authors can give the reasons on the model biases in the eastern boundary regions. It seems not directly related to resolution by comparing the results from the 23 CMIP6 models. 

Author Response

Reviewer 2:

This manuscript evaluated the simulated ability of the Eastern Boundary Upwelling System from 23 CMIP6 models, which is important for model result-users and model development. In general, the manuscript is well-structured and well-written. The method and conclusion are reliable. I think it can be accepted after clarifying several issues.

(a) As I know, the models participating in the CMIP6 are all general circulation models. Therefore, I do not think it’s necessary to include the “general circulation models” in the title. I suggest that the title can be changed to “On the ability of CMIP6 models to reproduce the upwelling SST imprint in the Eastern Boundary Upwelling Systems”.

This suggestion was considered in the new version of the manuscript

(b) It’s helpful if authors can give the meaning of “Historical”, “Hist-1950”, and “NICAM16-8S”. And I’m confused why the authors selected NICAM16-8S, which seems only an atmosphere model, as a control model.

This suggestion was considered in the new version of the manuscript (Lines 128-133).

Regarding to the NICAM16-8S, we selected it as a control model due to its particular characteristics. As an atmospheric model, the SST data for the NICAM16-8S was obtained from the daily quarter-degree sea surface temperature (SST) and sea ice mass (ICE) prescribed for the model were obtained from HadISST 2.2.0.0 (Kennedy et al., 2017). Then, the SST dataset is extended from 2016 to 2050 using a trend obtained from a CMIP5 model ensemble mean following the RCP8.5 scenario and historic variability from 1980 to 2015 (Kennedy et al., 2019). Thus, at last, we are comparing the results from OISST and HadISST in the case of this model. More information about the particularities of NICAM16-8S can be found at Kodama et al., (2020).

(c) As we know, there are several ensemble members in the CMIP6 published data, such as r1i1p1f1, r2i1p1f1, etc. In general, the climatological difference between r1 and r2 is very small, but it will be large between p1 and p2, or f1 and f2. The authors should clarify which ensemble members are used in this work.

We agree with the reviewer. Thus, in the new version of the manuscript we introduce this information in the Table 1.

We are aware of the differences between model configurations (variant label in this case). For example, in the case of CNRM-HR which is r1i1p1f2, following their own information: “given our simulations starting date, we used a version of forcing dataset not earlier than 6.2.0. That’s why we set forcing index=2, according to the CMIP6 guidelines at this date. Other modelling groups use a different number for the forcing index (commonly « f1 »). Note that this does not necessarily means that forcing version is different from our. Since the CMIP6 forcing version is unfortunately not documented by the CMIP6 file attributes, the only way to know is to ask the modelling centers or refer to the ES-DOC external model documentation (when available)”. (http://www.umr-cnrm.fr/cmip6/spip.php?article23).

In our particular case, we decided to use all the CMIP6 GCMs available in the moment we start this study with a nominal resolution less than 50km. As we study the models individually, if there were important differences due the variant label, they could be easily observed. Moreover, previous authors who conducted similar papers, also used all models available independently of their configurations (Song et al., 2020; Farneti et al., 2022; Liu et al., 2022; Sylla et al., 2022; Wang et al., 2022).

(d) It’ll be better if authors use the model’s name, even an abbreviation, instead of the number (such as 1, 2, 3, etc.).

This suggestion was considered in the new version of the manuscript.

(e) This year, Liu et al. (doi:10.1016/j.dsr2.2022.105120) evaluated the global SST simulated ability from the 48 CMIP6 models and showed the ensemble mean biases. As I know, it’s the result of the most number of CMIP6 models. Authors can compare with figure 1 of Liu’s article, although it only gives a global distribution.

We have extended the discussion section including this new reference (Lines 327-331).

(f) It’s not necessary to be answered, but it’s appreciated if authors can give the reasons on the model biases in the eastern boundary regions. It seems not directly related to resolution by comparing the results from the 23 CMIP6 models.

We agree with the reviewer that a discussion on the possible causes of the SST biases could be of interest. Thus, we have introduced a new paragraph (Lines 336-362) to discuss this topic.

References:

Kennedy, J., Titchner, H., Rayner, N. and Roberts, M.: input4MIPs.MOHC.SSTsAndSeaIce.HighResMIP.MOHC-HadISST2-2-0-0-0, Version 20170201, Earth System Grid Federation., 2017.

Kennedy, J., Titchner, H., Rayner, N. and Roberts, M.: input4MIPs.CMIP6.HighResMIP.MOHC.MOHC-highresSST future-1-0-1, Version 20190215., 2019.

Kodama, C., Ohno, T., Seiki, T., Yashiro, H., Noda, A. T., Nakano, M., ... & Sugi, M. (2020). The non-hydrostatic global atmospheric model for CMIP6 HighResMIP simulations (NICAM16-S): experimental design, model description, and sensitivity experiments. Geoscientific Model Development Discussions, 2020, 1-50.

Song, Z., Liu, H. and Chen, X. 2020. Eastern equatorial Pacific SST seasonal cycle in global climate models: from CMIP5 to CMIP6. Acta Oceanologica Sinica, 39(7), 50-60.

Farneti, R., Stiz, A. and Ssebandeke, J. B. 2022. Improvements and persistent biases in the southeast tropical Atlantic in CMIP models. npj Climate and Atmospheric Science, 5(1), 1-11.

Liu, H., Song, Z., Wang, X. and Misra, V. 2022. An ocean perspective on CMIP6 climate model evaluations. Deep Sea Research Part II: Topical Studies in Oceanography, 105120.

Sylla, A., Sanchez Gomez, E., Mignot, J. and López-Parages, J. 2022. Impact of increased resolution on the representation of the Canary upwelling system in climate models. Geoscientific Model Development Discussions, 1-35.

Wang, Y., Heywood, K. J., Stevens, D. P. and Damerell, G. M. 2022. Seasonal extrema of sea surface temperature in CMIP6 models. Ocean Science, 18(3), 839-855.

Round 2

Reviewer 1 Report

I suggest it to be accepted.

Reviewer 2 Report

The authors have addressed all my concerns. Now, I have no further comments.